# Towards Effective Updating of Pretrained Symbolic Music Models for Fine-Grained Bar-Level Control

## Abstract

Automatically generating symbolic music scores tailored to specific user needs offers significant benefits for musicians and enthusiasts alike. Pretrained symbolic music autoregressive models have demonstrated promising results, thanks to large datasets and advanced transformer architectures. However, in practice, the control provided by such models is often limited, particularly when fine-grained controls are needed at the level of individual bars. While fine-tuning the model with newly introduced control tokens may seem like a straightforward solution, our research reveals challenges in this approach, as the model frequently struggles to respond effectively to these precise bar-level control signals. To overcome this issue, we propose two novel strategies. First, we introduce a pre-training task that explicitly links control signals with their corresponding musical tokens, enabling a more effective initialization for fine-tuning. Second, we develop a unique counterfactual loss function that enhances alignment between the generated music and the specified control prompts. These combined methods substantially improve bar-level control, yielding a 13.06% improvement over the fine-tuning baseline. Importantly, subjective evaluations confirm that this increased control does not compromise the musical quality produced by the original pretrained model.

## 1 Introduction

Symbolic music generation, which focuses on the automatic creation of music scores, has gained significant attention in recent years due to its intuitive readability and excellent editability (Zhang et al., 2020; Wu & Sun, 2022). Autoregressive models using transformer architectures, such as Music Transformer (Anna et al., 2018), Museformer (Yu et al., 2022), and MuseCoco (Lu et al., 2023), have demonstrated promising results due to its impressive generation quality and scalability with large-scale datasets. Conditional music generation models, which allow for music to be generated based on user-defined attributes like style or rhythm, can also be integrated into autoregressive transformer frameworks (Wu & Sun, 2022; OpenAI, 2023; Lu et al., 2023). However, due to the many ways music generation can be controlled, it is challenging to account for all potential conditions during the training phase for large models. This raises the question of how a model can be quickly updated with new control conditions after it has already been trained on a large dataset.

In particular, granular control over music generation is highly desirable but remains lacking in existing pretrained autoregressive music models, such as (Lu et al., 2023). These models have predominantly focused on generating music based on broad, high-level descriptions, offering limited control over elements like tempo and style within individual sections of a composition. This lack of fine-grained control at the bar level [1] restricts users from making detailed adjustments to specific musical elements. Introducing bar-level control would provide users with greater creative flexibility and enhance applications in automatic composition. For example, attributes from a particular bar could be extracted and applied to generate new pieces, enabling style imitation. This level of control would also improve the alignment between lyrics and melody, ensuring the music conveys the desired

---

[1] While this paper focuses on "bar-level" control, the method can naturally be extended to other time units, such as "beat-level" control.

emotional cues. Moreover, identifying and utilizing attributes from favored music pieces to generate new compositions through bar-level manipulation would enable more customized and personalized music creation.

When a pretrained music generation model is available, a straightforward approach to incorporating bar-level control is to fine-tune the model with newly introduced control signals. Ideally, this fine-tuning should be performed using a relatively small dataset compared to the volume of the original training data for the foundation model, thereby avoiding the need for complete retraining. Specifically, we can utilize bar-level music attributes extracted from the training set's music scores as prefix control prompts to train an autoregressive model, with the objective of optimizing the likelihood of the training samples. However, we've found that models trained in this manner often fail to adhere to the guidance of the bar-level attributes. Our analysis suggests that the model struggles with interpreting the meanings of these new control prompts, leading to music that does not accurately align with these prompts. Furthermore, when the training data is limited, the model is prone to overfitting, focusing more on minimizing loss rather than effectively using the control prompt to steer music generation.

To address this limitation, we propose two strategies to improve bar-level controllability. The first strategy involves pretraining the control prompt and fine-tuning the model on an auxiliary task designed to promote accurate alignment between the control prompts and the generated music tokens. The second strategy introduces a counterfactual loss that penalizes the model for neglecting the bar-level guidance. By implementing these two techniques together, we significantly enhance the accuracy of bar-level control without compromising the quality of the music produced.

In sum, the key contributions of this work are outlined as follows:

- We conducted the first study in achieving fine-grained control of symbolic music generation based on the existing foundation model.

- We propose two innovative strategies, auxiliary task pre-training and counterfactual loss, to improve bar-level control in the foundation model.

## 2 RELATED WORK

### 2.1 TEXT-DRIVEN MUSIC GENERATION

Text-driven music generation, aimed at creating high-quality music from textual descriptions, has attracted considerable attention from researchers due to its user-friendly editing capabilities. However, the scarcity of paired text-music data presents a major challenge. To address this, some studies have employed diffusion-based methods (Zhu et al., 2023; Schneider et al., 2023) with self-collected text-audio datasets to facilitate text-audio music generation. MuLan (Huang et al., 2022) tackles the issue of data scarcity. They use techniques similar to those in CLIP (Radford et al., 2021) to contrastively embed two modalities: music pieces and their textual annotations. Building on this, MusicLM (Agostinelli et al., 2023) generates audio from MuLan's embeddings (Huang et al., 2022), enabling text-to-music conversion without the need for paired data. However, MusicLM's process of sampling to acquire fine-grained acoustic tokens is computationally intensive. Other efforts, like MeLoDy (Lam et al., 2024), seek to simplify music generation by efficiently translating conditioning tokens into sound waves. Furthermore, MusicGen (Copet et al., 2024) introduces a single-stage transformer LM framework that models multiple streams of acoustic tokens in parallel. Despite significant advancements in text-driven music generation, the methods are still relatively crude, limiting users' ability to edit musical elements within the generated audio. The controllability and editability of the outputs remain constrained.

### 2.2 SYMBOLIC MUSIC GENERATION

Compared to text-driven music generation, symbolic music provides easier editing capabilities, allowing users to manipulate specific musical elements more effectively. The development of solutions for this task has evolved significantly, from grammar rules-based methods (Collins, 2009; García Salas et al., 2011; Fernández & Vico, 2013) to probabilistic models and evolutionary computation (Yanchenko & Mukherjee, 2017; Liu & Ting, 2016), and more recently to neural networks and deep learning (Ycart et al., 2017; Briot et al., 2017; Dong et al., 2018). The advent of transformer-based

models, known for their successes in text generation (Li et al., 2022; Ren & Liu, 2023), has also influenced music generation. The Music Transformer (Anna et al., 2018) utilizes transformers with relative attention, proving highly effective for generating symbolic music. Museformer (Yu et al., 2022) addresses challenges in long-sequence and music structure modelling by capturing music structure-related correlations, thus enhancing music generation efficiency. The recently introduced MuseCoco (Lu et al., 2023) offers precise control over symbolic music generation through specific attributes, using these attributes as a bridge to transition from text-to-music to attribute-to-music generation. MuseCoco enables the adjustment of various musical attributes, offering a level of control akin to the compositional process. However, this control is limited to entire compositions, diverging from a composer's typical approach, which often involves more granular control, such as bar-level manipulation. Recent studies have explored the application of diffusion models to symbolic music generation, inspired by their success in the image and audio domains. Lv et al. (2023) introduced a framework called GETMusic, which includes a novel music representation, "GETScore," and a diffusion model, "GETDiff." The diffusion model is trained to predict masked target tokens conditioned on the source tracks. Similarly, Min et al. (2023) proposed a diffusion model, Polyffusion, for generating polyphonic music scores by treating music as an image-like piano roll representation. Their method enables controllable music generation, allowing users to pre-define parts of the music and apply external controls, such as chords or other musical features. Wang et al. (2024) explored the training of a cascaded diffusion model to capture hierarchical language structures, enabling whole-song hierarchical generation. While diffusion models offer a promising direction for symbolic music generation, the currently available pretrained models remain autoregressive. This paper will thus focus on updating autoregressive models with newly introduced control signals.

## 3 PRELIMINARIES

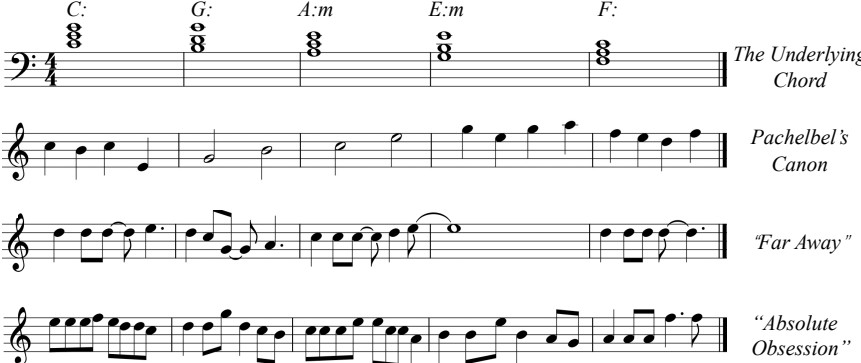

Figure 1: An example of three pop songs sharing the same chord progression. The top row displays five-column chords, while the bottom three rows represent the pop songs "Canon" by Pachelbel, "Far Away" by Jay Chou, and "Absolute Obsession" by Sam Lee.

We begin by examining the MuseCoco model (Lu et al., 2023), which serves as the foundation for our method. MuseCoco is a text-to-music generative model that initially converts text instructions into a set of music attributes and then generates music tokens based on these attributes. Our approach builds upon this attribute-to-music generation model.

In this model, a series of prefix tokens $\mathbf{x} = [x_1, x_2, ..., x_m]$ encodes the music attributes. Subsequently, the model generates a sequence of music tokens $\mathbf{y} = [y_1, y_2, ..., y_n]$. For additional details on the tokenization design, we refer to (Lu et al., 2023). The model is trained to maximize the log-likelihood of the ground-truth music sequences, as in the standard autoregressive model, namely:

$$\mathcal{L} = -\sum_{i=1}^{n} \log p(y_i | y_{<i}, x_{1:m}), \tag{1}$$

where $y_{<i}$ indicates historic tokens before $i$.

One limitation of the MuseCoco model (Lu et al., 2023) is its focus solely on global music attributes, which describe the overall composition without supporting finer control, such as at the bar level.

Fine-grained control is particularly valuable to both musicians and amateurs as it enables users to define specific properties for smaller segments of music. For instance, users can specify the chords[2] in each bar and provide a chord progression, and then explore various musical outputs based on that progression. Figure 1 illustrates how different music pieces can share the same chord progression. Beyond its utility for human users, fine-grained bar-level control is also advantageous for automated composition. For example, bar-level music attributes from one piece could be used to conditionally regenerate another piece, facilitating style mimicry. Additionally, this level of control can help achieve a closer alignment between the emotional content of lyrics and the corresponding melody, such as matching an intensification of emotion in the lyrics with an ascending melody line.

## 4 METHOD

To attain precise control, we present ControlMuse. This model overcomes the constraints of existing music score generation models that largely produce music based on broad and vague descriptions. Our method consists of three components: (1) we refine the control prompt in MuseCoco (Lu et al., 2023) to facilitate bar-level control instead of global control. (2) we incorporate an auxiliary task to pre-condition the model and the newly implemented control prompts. (3) we introduce a counterfactual loss to enhance the adherence of the generated music to the bar-level control prompts.

### 4.1 CONTROL PROMPT AUGMENTATION

MuseCoco (Lu et al., 2023) only incorporates global musical attributes that define the overall character of the music, which cannot achieve fine-grained bar-level control of music generation. To facilitate the latter, we first introduce a scheme to specify the bar-level music attribute. Specifically, ControlMuse processes a sequence of music attribute tokens $X = X_g, X_1, X_2, ...X_b$ as input, where $X_g = x_{g,1}, x_{g,2}, ...x_{g,|X_g|}$ comprises $|X_g|$ tokens representing global attributes, and $b$ denotes the number of music bars. Each $i$th bar $X_i = x_{i,1}, x_{i,2}, ..., x_{i,|X_i|}$ contains $|X_i|$ tokens representing the attributes at the bar level. For instance, token 24 in the chord attribute may indicate that the current bar's chord is "Eb: ", while token 0 in the rhythm intensity attribute could denote a "serene" rhythm (see Table 6 in Appendix). Those attributes can be extracted from the training music scores, i.e., from $y_1, y_2, ..., y_n$. For example, there are existing algorithms [3] to extract the chord implies in a given bar based on the distribution of the note pitches in the bar, whereas rhythm intensity is determined by the note density within the bar. Position embeddings are added to the bar-level tokens within this sequence $X_1, X_2, ...X_b$, distinguishing each bar's tokens shown in Figure 2. Tokens corresponding to the same bar are assigned identical position embeddings. Conversely, position embeddings are not used for the global tokens $X_g$. As a result, the input sequence in our method is encoded as follows:

$$X_g, X_1, X_2, ..., X_b, [SEP], y_1, y_2, ..., y_n. \qquad (2)$$

A straightforward approach involves fine-tuning the MuseCoco model by optimizing the likelihood of the ground-truth music sequence given the bar-level attributes derived from that same sequence, say:

$$\mathcal{L}_{\mathcal{BFT}} = -\sum_{i=1}^{n} \log p(y_i | y_{<i}, X), \qquad (3)$$

where $\mathcal{L}_{\mathcal{BFT}}$ denotes the **B**ar-level **F**ine-**T**uning loss. In practice, this process can be easily implemented via efficient parameter fine-tuning, such as LoRA (Hu et al., 2021). However, as detailed in the experimental section (see Table 1), this method proves to be less effective. We find that the model tends to overlook the newly introduced control prompts in its efforts to maximize the likelihood. Consequently, although the loss decreases, the controllability of the new prompts does not improve. We hypothesize that this issue arises because the new control prompts are randomly initialized and

---

[2]Chords are groups of notes played together that act as the foundation of music. They generate harmony and shape the emotional atmosphere of a piece. A sequence of changing chords, known as a chord progression, provides music with its rhythm and supports the melody. Often in music, especially in piano compositions, chords manifest not through simultaneous notes but through sequentially played notes, typically with the left hand, known as a broken chord. The specific chords used in a piece can be identified by analyzing the notes that appear throughout the composition.

[3]We use the algorithm provided by https://github.com/Rainbow-Dreamer/musicpy.

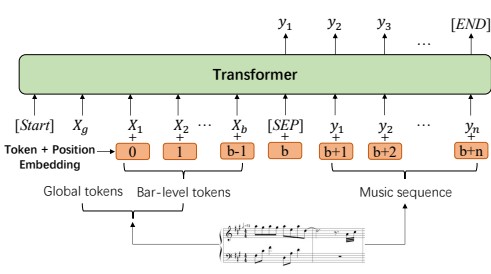

Figure 2: Control prompt augmentation.

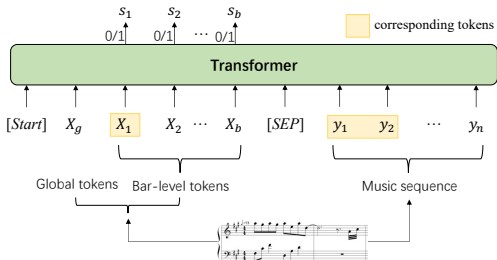

Figure 3: Pre-adapt the bar-level control.

they have not been supported by the foundation. As a result, the MuseCoco model struggles to adequately translate these controls into the music generation process. Furthermore, the model can more readily maximize likelihood by overfitting the data, such as by memorizing the training pieces, thereby diminishing any incentive for the network to adhere to the control signals.

To address this issue, this paper proposes two strategies to improve the controllability of the network.

### 4.2 CONTROL PROMPTS PRE-ADAPTATION VIA AN AUXILIARY TASK (PA)

The first strategy is to utilize an auxiliary task to pre-adapt the bar-level control mechanism of the MuseCoco model. In this context, "pre-adapt" refers to modifying the embeddings of the control prompts and the LoRA parameters. The auxiliary task is framed to meet two key criteria: firstly, it must be closely related to the task of bar-level controlled generation, ensuring that the parameters refined during the auxiliary task can be effectively transferred to the main generation task. Secondly, the model can only accomplish the auxiliary task by genuinely utilizing the interactions between the control prompts and the corresponding music tokens. Specifically, to achieve bar-level controlled generation, it is essential for the model to comprehend which music tokens and specific attributes of those tokens are governed by a bar-level prompt. Accordingly, we propose a recognition task: the model is presented with a sequence of bar-level control prompts alongside a sequence of music tokens, and it must determine whether the music sequence conforms to the guidelines set by the bar-level control prompts. For this purpose, we develop new linear projection heads that are trained on the output embeddings of the bar-level control prompts. These projections perform a binary classification task to assess whether the specified bar-level control, as indicated by the control prompt, has been adhered to in the music sequence. This setup is depicted in Figure 3. Successfully completing this task hinges on the accurate correlation between the control prompts and the music tokens, a skill that is directly transferable to controlled music generation.

In our implementation, we randomly select the number of $t$ bars, denoted as $\mathcal{M}$ and $|\mathcal{M}| = t$, and modify the corresponding bar-level tokens in the input to reflect that the attributes of these bars do not match the expected values. Then we create a sequence of ground-truth prediction labels $\{s_i\}$, with $s_i = 1$ for unmodified bar and $s_i = 0$ for modified bars ("1" indicates "match" and "0" indicates "unmatch"). The training loss for this step is expressed as follows:

$$\mathcal{L}_{\mathcal{PA}} = -\sum_{i \in \mathcal{M}} \log(1 - p(s_i|\mathbf{y}, X')) - \sum_{j \notin \mathcal{M}} \log(p(s_j|\mathbf{y}, X')), \tag{4}$$

where $X'$ represents the set of all bar-level attribute tokens, with modifications made to some bars, such as changing the chord from C: to A:m or changing the rhythm intensity from serene to moderate.

### 4.3 IMPROVING CONTROLLABILITY VIA COUNTERFACTUAL LOSS (CF)

The second strategy incorporates the use of a counterfactual loss to verify that the generated tokens are genuinely influenced by the bar-level control prompts. Our rationale is that if the music tokens of a particular bar are truly driven by its associated bar-level control prompt, then altering the control prompt should result in a substantial decrease in the likelihood of those specific music tokens. Specifically, we randomly replace the bar-level attributes $X_i \in \{X_1, X_2, ..., X_b\}$ with a different value within the attribute, denoted as $\overline{X_i}$. We then measure the change of the

negative log-likelihood of the $i$th bar's token, represented by the difference $\mathcal{J}_2 - \mathcal{J}_1$, where $\mathcal{J}_1 = -\frac{1}{|bar_i|} \sum_{i \in bar_i} \log p(y_i|y_{<i}, X_g, X_1, ..., X_i, ..., X_b)$ is the negative log-likelihood before the change in this bar, and $\mathcal{J}_2 = -\frac{1}{|bar_i|} \sum_{i \in bar_i} \log p(y_i|y_{<i}, X_g, X_1, ..., \overline{X_i}, ..., X_b)$ indicates the negative log-likelihood after the change in this bar. $\overline{X_i}$ represents a randomly selected attribute token. In our implementation, to enhance the model's recognition of the governance range of bar-level attributes, we assign $\overline{X_i}$ the same value as $X_{i-1}$ whenever $X_i \neq X_{i-1}$.

This approach is designed to reinforce the model's awareness of the influence exerted by bar-level attributes. The counterfactual loss is thus defined as:

$$\mathcal{L}_{\mathcal{CF}} = max\{0, \eta - (\mathcal{J}_2 - \mathcal{J}_1)\}, \tag{5}$$

where the counterfactual loss $\mathcal{L}_{\mathcal{CF}}$ is designed to promote a significant decrease in the log-likelihood when the alignment between the control prompt and the music tokens is disrupted after modifying $X_i$. $\eta$ is a margin parameter specifying the desired log-likelihood change.

Overall, our method first pre-adapts the model with the training task described in Section 4.2 and then apply the counterfactual loss together with the bar-level fine-tuning loss:

$$\mathcal{L} = \mathcal{L}_{\mathcal{BFT}} + \lambda\mathcal{L}_{\mathcal{CF}}, \tag{6}$$

where $\lambda$ is the trade-off hyperparameter.

### 4.4 INFERENCE

During the inference stage, music sequences are generated on a bar-by-bar basis. Within each bar, up to $K$ sampling attempts are allowed. If a sample accurately reflects the intended bar-level attributes, the generation continues from the subsequent bar. Conversely, if $K$ samples all fail to exhibit the correct attributes, the token with the highest probability is chosen to continue the sequence prediction. When $K = 1$, this process simplifies to a typical auto-regressive model sampling procedure. Generally, a larger $K$ enhances controllability at the expense of reduced sampling efficiency. It is noteworthy that in many cases, the model may produce valid music on the first attempt, implying that the actual number of sampling operations may increase sublinearly with $K$.

## 5 EXPERIMENTS

In this section, we assess our method through two case studies focused on bar-level chord control and rhythm intensity control. In other words, the bar-level attribute corresponds to the chord and rhythm intensity for each bar, respectively. Bar-level chord control is useful as in music composition, particularly in pop music, it is customary to first establish a chord progression pattern before composing the music. Similarly, bar-level rhythm intensity control is beneficial for piano learners, particularly beginners, as it allows for the adjustment of note density to modify the difficulty of the generated music. It's important to note that the objective of our experiment is to evaluate the effectiveness of the proposed methods in enhancing control based on a pre-trained music model. We are not aiming to optimize performance specifically for chord-controlled and rhythm-intensity-controlled generation.

### 5.1 EXPERIMENTAL SETTING

#### 5.1.1 DATASETS

In our study, we use the POP909 dataset (Wang et al., 2020) to train and evaluate the proposed method. This dataset comprises multiple renditions of the piano arrangements for 909 popular songs, totalling approximately 60 hours of music. These arrangements are meticulously crafted by professional musicians and are provided in MIDI format. On average, each song consists of 270 bars. It is important to note that, in comparison to the training data used for MuseCoco, POP909 represents only a small subset. We chose this dataset for two reasons: (1) Pop music typically features distinct chord progression patterns, making it ideal for this study. (2) Pop piano music is more accessible for general audiences to evaluate its quality. Following (Lu et al., 2023), we randomly selected three 16-bar clips from each MIDI file. From these clips, global attributes are extracted from the entire clip,

while bar-level attributes—chords and rhythm intensity are extracted from each individual bar within the 16-bar length clip. The specific predefined musical attribute values, including global attributes sourced from the MuseCoco project (Lu et al., 2023) and chord and rhythm intensity attributes derived from MIDI files, are displayed in Appendix A.1. Details regarding the distribution of chords within this dataset are provided in Appendix A.3. To convert the MIDI files into token sequences, we employ a REMI-like representation method (Huang & Yang, 2020). For training, validation, and testing purposes, we partition the songs into three sets, with a split ratio of 8:1:1 for training, validation, and testing, respectively.

### 5.1.2 IMPLEMENTATION DETAILS

We employ the Linear Transformer architecture as our backbone model (Katharopoulos et al., 2020), configured with a causal attention mechanism spanning 24 layers and utilizing 24 attention heads. The hidden size is set to 2048, while the feedforward network (FFN) hidden size is 8192. During the training phase, we began by initializing the MuseCoco-base weights (Lu et al., 2023) with fp16 precision and subsequently applied a fine-tuning approach. Within the attention layers, LoRA Adapters (Hu et al., 2021) was incorporated, with a rank size of $r = 8$. The maximum sequence length was set to 5120. We use validation performance to set the margin $\eta$ to 0.05. To execute the fine-tuning process, we utilized 4 40GB-A100 GPUs, conducting 50 epochs for the first strategy (PA), and 40 epochs for the second strategy (BFT and CF). We utilize a batch size of 4 and employ the Adam optimizer (Kingma & Ba, 2014) with a learning rate of $2 \times 10^{-4}$, incorporating a warm-up step of 16,000 and an invert-square-root decay schedule. For inference, we consider the top 15 highest probabilities as potential prediction hypotheses and perform sampling $K = 15$ times.

### 5.1.3 COMPARED MODELS

In this investigation, we conduct a comparative analysis between our proposed method and **MuseCoco** (Lu et al., 2023) for symbolic music generation. Since MuseCoco is only fed with global musical attributes, we examine MuseCoco with **B**ar-level **F**ine-**T**uning, represented as **BFT** which builds upon the MuseCoco model by incorporating bar-level training techniques. This approach involves aligning the music attributes with each bar chord or rhythm intensity in the input.

We also conduct a comparative analysis between our proposed method and other Transformer-based conditional models, specifically **FIGARO** (von Rütte et al., 2023), as well as the diffusion-based model **Polyffusion** (Min et al., 2023). FIGARO employs a description-to-sequence learning approach, reconstructing the original sequence from fine-grained control signals such as chords and instrument information. Polyffusion, on the other hand, treats music as image-like piano roll representations and utilizes external control signals, such as chords, to achieve controllable music generation.

### 5.1.4 OBJECTIVE EVALUATION

- **Bar-level Attribute Accuracy:** We employ chord accuracy and rhythm intensity accuracy to assess the alignment between the prompted chords and rhythm intensity, and those generated during the symbolic music generation process. This evaluation offers insight into the bar-level controllability achieved by the proposed method.
- **Global Attribute Accuracy:** We use global attribute accuracy to evaluate the alignment between prompted global musical attributes and those generated during the inference process. This metric offers insight into the sample-level controllability of our method.

### 5.1.5 SUBJECTIVE EVALUATION

- **Musicality:** measures the extent to which generated music resembles music created by humans.

We randomly selected 20 pieces of music generated by MuseCoco performed with piano and our model and created a survey (see Figure 9 in Appendix). We then enlisted 16 piano teachers to evaluate which pieces resembled human-created music more closely. The survey options were 'Music 1', 'Music 2', or 'Similar', where Music 1 and Music 2 were randomly assigned to either the MuseCoco generation or ours [4]. The degree of musicality was gauged based on the percentage of votes received for each option.

---

[4]We have included the 20 pieces in the Supplementary Material and converted the MIDI files to MP3 format.

Table 1: Comparative analysis (%) of objective and subjective evaluations among MuseCoco, BFT, and ControlMuse.

| Method | Musicality | Chord Acc. | Rhythm Intensity Acc. | Avg. Global Attribute Acc. |
|---|---|---|---|---|
| MuseCoco | 40.63 | - | - | 78.14 |
| BFT | - | 65.27 | 76.37 | 81.54 |
| ControlMuse | **43.75** | **78.33** | **85.52** | **82.55** |

Table 2: Comparative analysis of chord accuracy and rhythm intensity accuracy among FIGARO, Polyffusion and ControlMuse.

| Method | Chord Accuracy (%) | Rhythm Intensity Accuracy (%) |
|---|---|---|
| FIGARO (von Rütte et al., 2023) | 70.06 | 80.33 |
| Polyffusion (Min et al., 2023) | 36.38 | - |
| ControlMuse | **78.33** | **85.52** |

## 5.2 COMPARED WITH MUSECOCO

Table 1 presents the results of both objective and subjective evaluations. Chord accuracy and average global attribute accuracy are calculated over $K = 15$ sampling iterations, whereas rhythm intensity accuracy is evaluated based on $K = 1$ sampling iteration. The findings are as follows: 1) In terms of musicality, ControlMuse achieves performance comparable to MuseCoco, with scores of 43.75% versus 40.63% respectively (the full survey statistic is shown in Figure 4.). This equivalence in scores demonstrates that ControlMuse preserves the musicality inherent in MuseCoco. 2) Regarding chord accuracy and rhythm intensity accuracy, MuseCoco lacks the capability to generate specific chords and rhythm intensity aligned with each bar. Compared to BFT, ControlMuse significantly improves chord accuracy and rhythm intensity accuracy by 13.06% and 9.15%, underscoring its enhanced controllability at the bar level. 3) Concerning average global attribute accuracy, ControlMuse outperforms BFT by a slight margin of 1.01%, and both show a substantial improvement over MuseCoco. The data indicates that aligning music attributes with each bar in the input significantly enhances global attribute accuracy. This improvement suggests that the effective generation of bar-level attributes can positively influence global attribute generation.

## 5.3 COMPARED WITH OTHER TRANSFORMER-BASED CONDITIONAL MODELS AND DIFFUSION MODELS

In this section, we compare the proposed method with the Transformer-based model FIGARO and the diffusion-based model Polyffusion, both of which utilize chord or rhythm intensity as fine-grained control signals. The comparative results are presented in Table 2. The proposed method achieves the highest accuracy in both chord and rhythm intensity, underscoring its efficacy in accurately responding to fine-grained bar-level control signals. This result further indicates the method's capability to effectively integrate additional control through fine-tuning the pre-trained music model, enabling it to respond proficiently to newly introduced, precise control signals.

## 5.4 ABLATION STUDY

### 5.4.1 COMPONENT-WISE ANALYSIS

In this section, we conduct ablation studies to evaluate the impact of each component within our method. The findings are detailed in Table 3. From the table, it can be seen that employing BFT alone only yields a chord accuracy of 65.27% and rhythm intensity accuracy of 76.37% during the inference stage. Notably, the inclusion of PA or CF significantly enhances performance, with improvements of 6.95% and 5.88% in chords, and 3.68% and 4.25% in rhythm intensity, respectively. This highlights the critical roles that PA and CF play in improving chord recognition and rhythm intensity and enhancing bar-level control. Furthermore, the simultaneous use of PA and CF leads to the best performance, achieving a substantial increase in chord accuracy from 72.22% to 78.33%

Table 3: Analyzing the impact of proposed components on chord accuracy and rhythm intensity accuracy: chord accuracy evaluated over $K = 15$ sampling iterations and rhythm intensity accuracy over $K = 1$ sampling iteration.

| BFT | PA | CF | Chord Accuracy (%) | Rhythm Intensity Accuracy (%) |
|:---:|:---:|:---:|:---:|:---:|
| ✓ | × | × | 65.27 | 76.37 |
| ✓ | ✓ | × | 72.22 | 80.05 |
| ✓ | × | ✓ | 71.15 | 80.62 |
| ✓ | ✓ | ✓ | **78.33** | **85.52** |

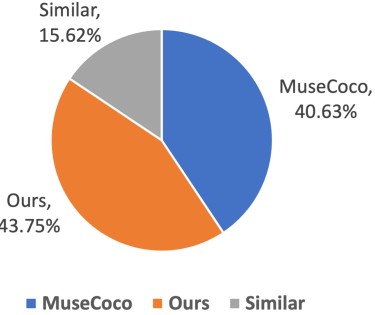

Figure 4: The vote percentages of music generated by MuseCoco and our method, as judged by 16 piano teachers for similarity to human creation.

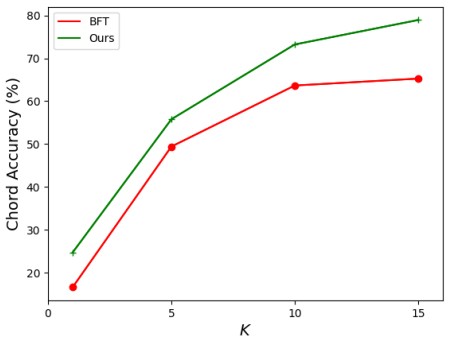

Figure 5: Chord accuracy of BFT and our method with respect to varying $K$ values ranging from 1 to 15.

with PA and from 71.15% to 78.33% with CF, Similarly, rhythm intensity accuracy improves from 80.05% to 85.52% with PA, and 80.62% to 85.52% with CF. These components enhance chord accuracy and rhythm intensity accuracy from distinct perspectives: PA aligns bar-level attributes and musical sequences as input, helping the model to effectively capture the relationships between bar-level attributes and corresponding music sequences; CF is designed to ensure the model correctly responds to the prefix bar-level prompts and avoids the trivial solution of generating the next tokens solely based on previous tokens.

### 5.4.2 IMPACT OF $K$ —THE NUMBER OF SAMPLING IN INFERENCE

In our method, during the inference phase, we select music sequences bar-by-bar from the $K$ sampling attempts. This section explores the impact of the value of $K$. Figure 5 displays the accuracy achieved with different $K$ values. The results indicate that as $K$, the number of sampling attempts, increases, chord accuracy rises significantly. Additionally, compared to BFT, ControlMuse consistently delivers superior performance across various $K$ values, with the performance gap between BFT and ControlMuse widening as $K$ increases.

### 5.4.3 IMPACT OF THE PARAMETER $\lambda$

To assess the impact of the parameter $\lambda$, we conducted experiments using varying values of $\lambda$. Table 4 presents the chord accuracy corresponding to five different settings of $\lambda$. As observed, when $\lambda$ is set to 0, the approach defaults to PA+BFT. As $\lambda$ increases from 0 to 1e3, there is a gradual improvement in chord accuracy, reaching a peak of 78.33% when $\lambda$ is approximately 1e3. Beyond this point, the performance begins to decline slightly.

### 5.5 COMPLEXITY ANALYSIS

Assuming the length of the music sequence is $n$ and the number of sampling attempts per bar is $K$, the time complexity of the inference step in MuseCoco is $O(n)$, whereas in our method it is $O(Kn)$. Although the time complexity scales with the number of sampling attempts $K$, the actual sampling often terminates earlier if the model successfully matches the bar-level attributes early on.

Table 4: Chord accuracy (%) of the proposed method with different $\lambda$.

| $\lambda$ | Chord Accuracy |
|---|---|
| 0 | 72.22 |
| 1e1 | 73.79 |
| 1e2 | 76.42 |
| 1e3 | **78.33** |
| 1e4 | 76.64 |

Table 5: The average inference time on MuseCoco and our method per sample. $K = 5$ and $K = 15$ denote the number of samples in inference. The inference times were measured on an NVIDIA RTX 4090 GPU.

| Method | Runtime |
|---|---|
| MuseCoco | 3 mins |
| Ours ($K = 5$) | 4 mins |
| Ours ($K = 15$) | 6 mins |

Consequently, the average number of samples taken at each bar is typically less than $K$. The average inference time per sample with bar-level attributes set to chord is presented in Table 5.

### 5.6 COMMENTS FROM MUSICIAN AND COMPOSER

We invited musicians to provide feedback on the music created by our system and composers to experience how our system aids the creative process. Their guidance and comments are as follows:

From Musician: "*I was truly impressed by the music produced by this system; its performance is remarkable. The quality of the music is very similar to that of human compositions, and some of the bar chord arrangements are astonishing*".

From Composer: "*This system for chord arrangement and music creation significantly reduces my composition time. I found it to be a great source of inspiration. For instance, when I wanted to arrange the next chord as an A:m chord, the system provided many options to choose from. However, it uses a lot of broken chords, which composers typically don't use as frequently in their compositions. It would be beneficial if this aspect could be improved in the future*".

## 6 LIMITATION AND FUTURE WORK

This work focuses on attribute-to-music generation, directly specifying attribute values to control the bar-level music generation process. However, this approach may not be user-friendly for those who prefer to use text descriptions for control. Therefore, in the future, we aim to develop a more user-friendly interface that allows bar-level music generation from text descriptions, enabling users to create and edit bars using natural language. Also, we plan to build a system incorporating more diverse control signals.

## 7 CONCLUSION

In this paper, we introduce ControlMuse, a method that allows for finer detail control at the level of individual bars, significantly advancing the field of automated music score composition and alignment. This innovative approach offers substantial value and potential for both musicians and amateurs, enhancing creative efficiency and providing greater control over the composition process. Our solution contains two innovative strategies that enhance bar-level control without compromising the quality of the music produced. These designs enable the model to accurately adapt to the adjusted bar-level attributes as new control prompts, thereby achieving impressive bar-level controllability.

Our research demonstrates the feasibility of bar-level editing in AI technologies. Successful results in chord control and generation, as shown in Appendix A.2, suggest that more bar-level attributes, such as melody trends, can be explored in the future. We hope that further inspiration for bar-level attributes will enhance the ability to edit and control bar generation, thereby improving music creation. It is possible to adopt ControlMuse for more bar-level attribute edits like melody trends. We hope ControlMuse will be even more useful with additional bar-level attributes, utilizing the proposed two training strategies.

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

# A APPENDIX

## A.1 PRE-DEFINED MUSICAL ATTRIBUTES

Table 6 displays the global attributes and their values, as well as the bar-level attributes and their values.

Table 6: Global and bar-level attribute value. "NA" indicates that the attribute is not mentioned in the text, while "N.C." denotes the absence of any chords.

| Type | Attribute | Values |
|---|---|---|
| Global | Instrument | piano. 0: played, 1: not played, 2:NA |
| | Pitch Range | 0-11: octaves, 12:NA |
| | Bar | 0: 1-4 bars, 1: 5-8 bars, 2: 9-12 bars, 3: 13-16 bars, 4: NA |
| | Time Signature | 0: 4/4, 1: 2/4, 2: 3/4, 3: 1/4, 4: 6/8, 5: 3/8, 6: others, 7: NA |
| | Key | 0: major, 1: minor, 2: NA |
| | Tempo | 0: slow (<=76 BPM), 1: moderato (76-120 BPM), 2: fast (>=120 BPM), 3: NA |
| | Time | 0: 0-15s, 1: 15-30s, 2: 30-45s, 3: 45-60s, 4: >60s, 5: NA |
| Bar-level | Chord | 0: C:, 1: C:m, 2: C:+, 3: C:dim, 4: C:7, 5: C:maj7, 6: C:m7, 7: C:m7b5, 8: C#:, 9: C#:m, 10: C#:+, 11: C#:dim, 12: C#:7, 13: C#:maj7, 14: C#:m7, 15: C#:m7b5, 16: D:, 17: D:m, 18: D:+, 19: D:dim, 20: D:7, 21: D:maj7, 22: D:m7, 23: D:m7b5, 24: Eb:, 25: Eb:m, 26: Eb:+, 27: Eb:dim, 28: Eb:7, 29: Eb:maj7, 30: Eb:m7, 31: Eb:m7b5, 32: E:, 33: E:m, 34: E:+, 35: E:dim, 36: E:7, 37: E:maj7, 38: E:m7, 39: E:m7b5, 40: F:, 41: F:m, 42: F:+, 43: F:dim, 44: F:7, 45: F:maj7, 46: F:m7, 47: F:m7b5, 48: F#:, 49: F#:m, 50: F#:+, 51: F#:dim, 52: F#:7, 53: F#:maj7, 54: F#:m7, 55: F#:m7b5, 56: G:, 57: G:m, 58: G:+, 59: G:dim, 60: G:7, 61: G:maj7, 62: G:m7, 63: G:m7b5, 64: Ab:, 65: Ab:m, 66: Ab:+, 67: Ab:dim, 68: Ab:7, 69: Ab:maj7, 70: Ab:m7, 71: Ab:m7b5, 72: A:, 73: A:m, 74: A:+, 75: A:dim, 76: A:7, 77: A:maj7, 78: A:m7, 79: A:m7b5, 80: Bb:, 81: Bb:m, 82: Bb:+, 83: Bb:dim, 84: Bb:7, 85: Bb:maj7, 86: Bb:m7, 87: Bb:m7b5, 88: B:, 89: B:m, 90: B:+, 91: B:dim, 92: B:7, 93: B:maj7, 94: B:m7, 95: B:m7b5, 96: N.C. |
| | Rhythm Intensity | 0: serene, 1: moderate, 2: intense, 3: NA |

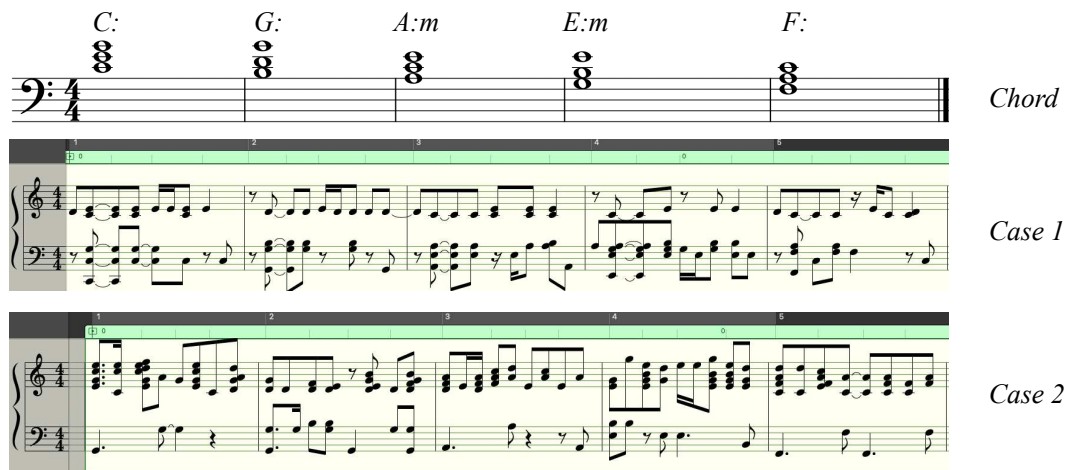

Figure 6: Two generated examples using "Canon chord progression" with different global attribute controls.

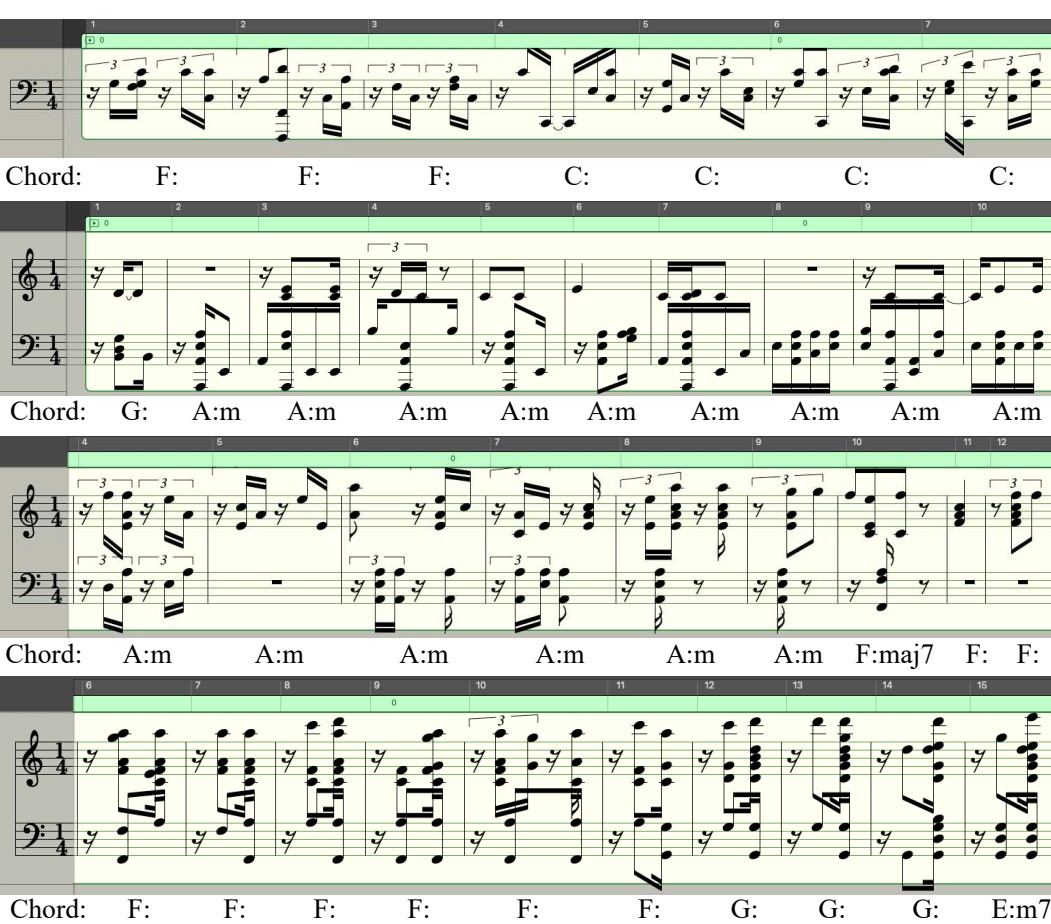

Figure 7: Some good cases of generated piano rolls, where the chords in the generated music perfectly align with the prompts.

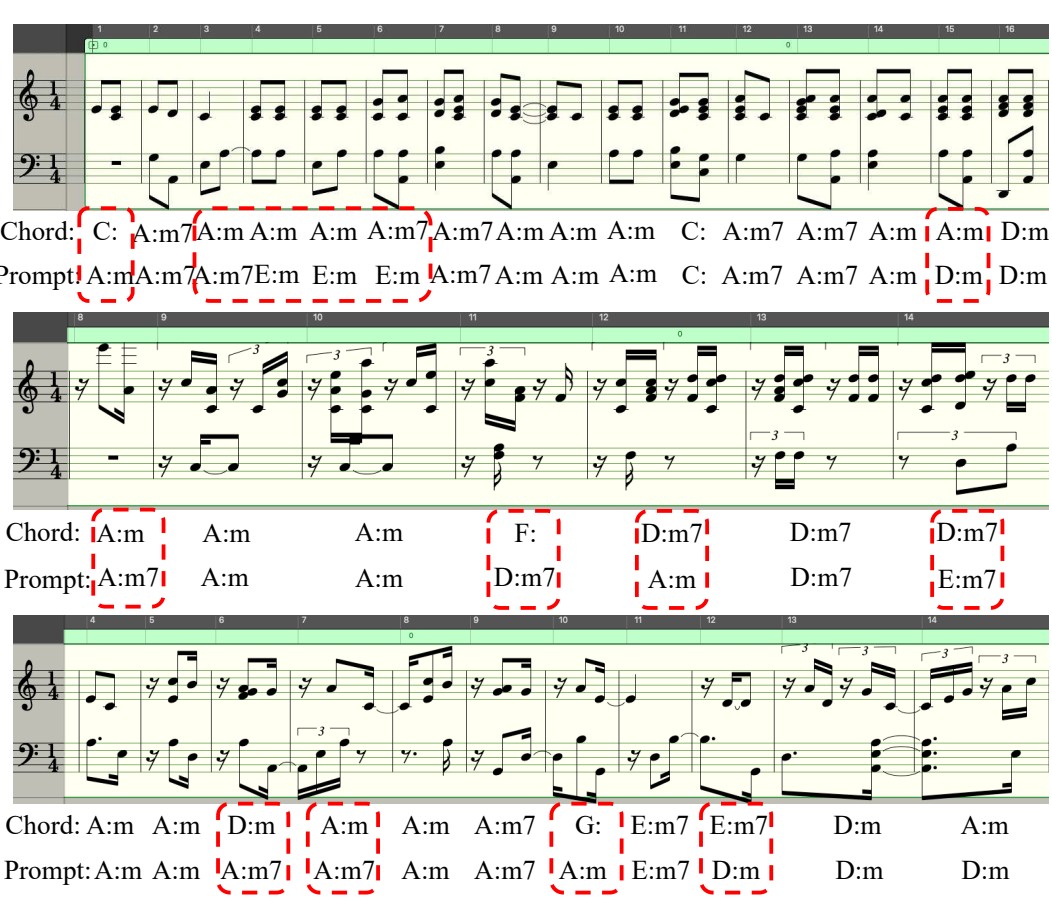

Figure 8: Examples of failure cases in generated piano rolls, where the chords in the music do not align with the prompts, highlighted by red boxes.

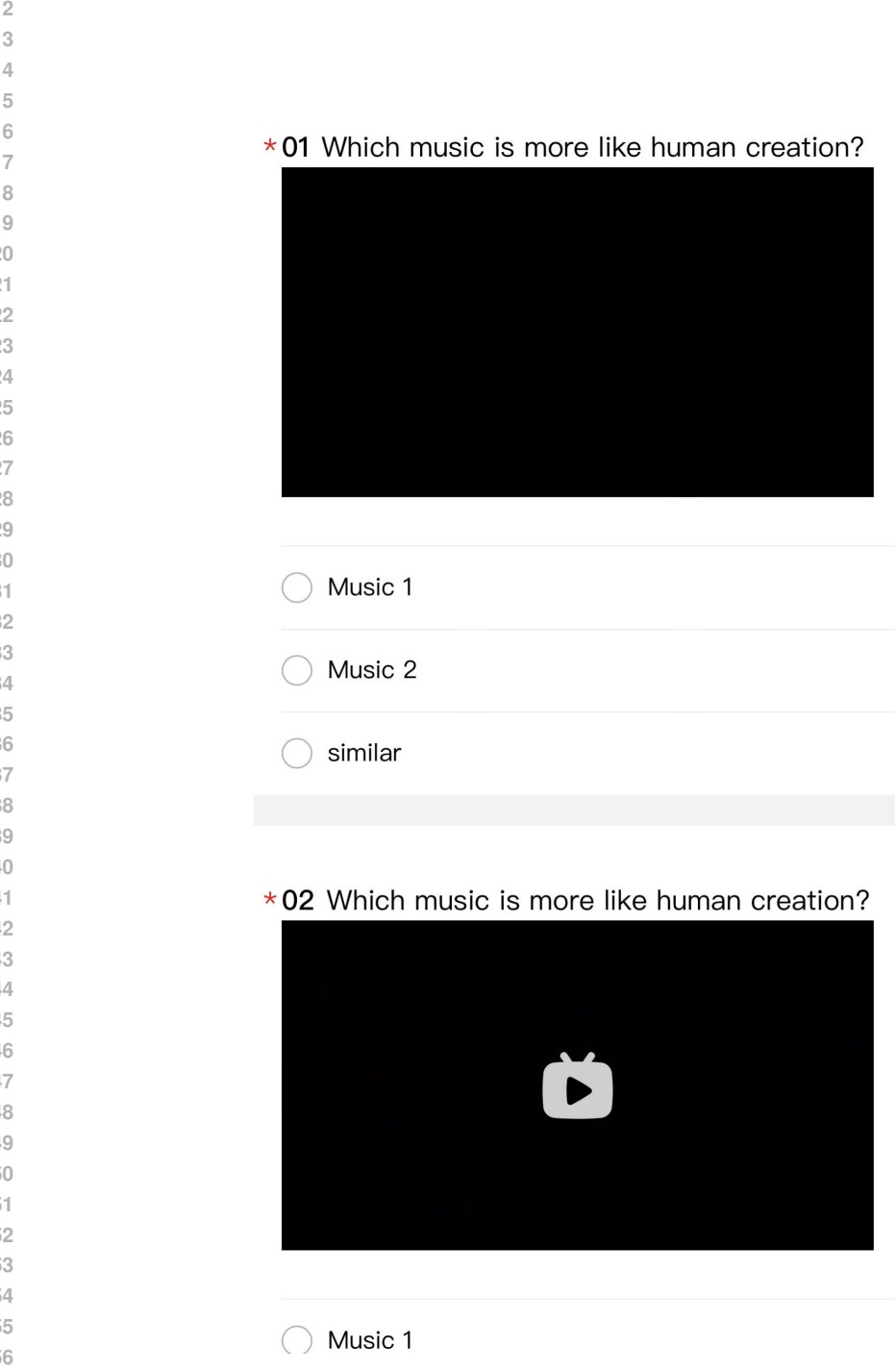

Figure 9: Screenshot of Musicality Judgment Data Collection: MuseCoco vs. Our Method.

## A.2 PIANO ROLLS ANALYSIS

### A.2.1 CANON-STYLE CASE GENERATION

We conducted experiments using the "Canon chord progression", as shown in Figure 1, to generate new melodies. In addition, we applied different global attributes to generate distinct melodies while maintaining the Canon style. The global attributes for the first case are 4 octaves, moderato tempo, and minor key. For the second case, the attributes are 2 octaves, fast tempo, and major key. The piano rolls for these two cases are displayed in Figure 6, and the audio can be found in the Supplementary Material. Interestingly, we found that both cases retain the Canon style, with melodies that are comfortable, beautiful, and distinctly Canon-like.

### A.2.2 GOOD CASES

We selected some successful cases from our testing where the generated chords perfectly align with the prompts at the bar level, as shown in Figure 7. This demonstrates the proposed method's strong ability to control each bar's generation with the correct chords.

### A.2.3 FAILURE CASES

We also selected some failure cases to analyze and observe how the model failed to generate the correct chords in each bar, as shown in Figure 8. In these instances, we found that some prompt chords, such as A:m7, were incorrectly generated as A:m, and D:m7 was incorrectly generated as F. Additionally, the model frequently confused D:m, D:m7, and E:m7. This indicates that the model sometimes struggles to differentiate between similar chords (e.g., A:m: A-C-E vs A:m7: A-C-E-G; D:m7: D-F-A-C vs F: F-A-C).

## A.3 CHORD PROPORTIONS.

The complete chord distribution is presented in Table 7. As shown, the common chords in the POP909 dataset include C, C:maj7, D, D:m, D:m7, E, E:m, E:m7, F, F:maj7, G, A:m, and A:m7. Some chords, however, never appear in the dataset, such as D:+, E:+, F:+, F:m7b5, F#:+, F#:7, G:+, Ab:+, Ab:m7b5, A:+, Bb:+, Bb:dim, Bb:m7b5, B:+, and B:maj7. Given that the selected dataset primarily focuses on popular music, it does not encompass all possible chords. Nonetheless, to ensure the scalability of our method, we uniformly model all chords.

## A.4 HUMAN EVALUATION

Figure 9 shows the voting interface.

Table 7: Statistics of chord proportions in the POP909 dataset. blue represents the chord and its proportion more than 1%. Since the selected dataset mainly focuses on popular music, it does not cover all possible chords. However, to ensure the scalability of the method, we uniformly model all chords.

| Chord | Proportion (%) | Chord | Proportion (%) | Chord | Proportion (%) |
|---|---|---|---|---|---|
| C: | 17.11 | E:m | 6.97 | Ab:+ | 0 |
| C:m | 0.13 | E:+ | 0 | Ab:dim | 0.04 |
| C:+ | 0.08 | E:dim | 0.05 | Ab:7 | 0.02 |
| C:dim | 0.01 | E:7 | 0.21 | Ab:maj7 | 0.06 |
| C:7 | 0.17 | E:maj7 | 0.01 | Ab:m7 | 0.01 |
| C:maj7 | 1.62 | E:m7 | 3.24 | Ab:m7b5 | 0 |
| C:m7 | 0.05 | E:m7b5 | 0.03 | A: | 0.63 |
| C:m7b5 | 0.01 | F: | 9.94 | A:m | 18.31 |
| C#: | 0.38 | F:m | 0.57 | A:+ | 0 |
| C#:m | 0.02 | F:+ | 0 | A:dim | 0.01 |
| C#:+ | 0.01 | F:dim | 0.01 | A:7 | 0.07 |
| C#:dim | 0.02 | F:7 | 0.02 | A:maj7 | 0.02 |
| C#:7 | 0.01 | F:maj7 | 2.25 | A:m7 | 3.91 |
| C#:maj7 | 0.06 | F:m7 | 0.12 | A:m7b5 | 0.01 |
| C#:m7 | 0.02 | F:m7b5 | 0 | Bb: | 0.52 |
| C#:m7b5 | 0.01 | F#: | 0.16 | Bb:m | 0.23 |
| D: | 1.18 | F#:m | 0.09 | Bb:+ | 0 |
| D:m | 8.03 | F#:+ | 0 | Bb:dim | 0 |
| D:+ | 0 | F#:dim | 0.20 | Bb:7 | 0.01 |
| D:dim | 0.06 | F#:7 | 0 | Bb:maj7 | 0.07 |
| D:7 | 0.18 | F#:maj7 | 0.04 | Bb:m7 | 0.08 |
| D:maj7 | 0.03 | F#:m7 | 0.05 | Bb:m7b5 | 0 |
| D:m7 | 3.02 | F#:m7b5 | 0.08 | B: | 0.07 |
| D:m7b5 | 0.09 | G: | 14.13 | B:m | 0.21 |
| Eb: | 0.16 | G:m | 0.17 | B:+ | 0 |
| Eb:m | 0.09 | G:+ | 0 | B:dim | 0.22 |
| Eb:+ | 0.02 | G:dim | 0.01 | B:7 | 0.01 |
| Eb:dim | 0.01 | G:7 | 0.98 | B:maj7 | 0 |
| Eb:7 | 0.01 | G:maj7 | 0.08 | B:m7 | 0.12 |
| Eb:maj7 | 0.03 | G:m7 | 0.12 | B:m7b5 | 0.48 |
| Eb:m7 | 0.05 | G:m7b5 | 0.01 | N.C. | 0.37 |
| Eb:m7b5 | 0.01 | Ab: | 0.45 | | |
| E: | 1.84 | Ab:m | 0.01 | | |

