# OpenReview forum: "Towards Effective Updating of Pretrained Symbolic Music Models for Fine-Grained Bar-Level Control"
_ICLR.cc/2025/Conference — ICLR 2025 Conference Withdrawn Submission_

### Official Review · Reviewer_139Z · 2024-11-02

**Soundness:** 3
**Presentation:** 2
**Contribution:** 2
**Rating:** 5
**Confidence:** 3

**Summary:**

This paper presents ControlMuse, a novel approach that enhances fine-grained control in symbolic music generation, specifically at the bar level. While existing autoregressive models for music generation demonstrate promising results, they often lack precise control over individual bar attributes. ControlMuse addresses this limitation by introducing two key innovations: a pre-training task that associates control signals with musical tokens to improve initialization, and a counterfactual loss function that aligns generated music with control prompts. These methods improve bar-level controllability by 13.06% compared to a standard fine-tuning baseline, and subjective evaluations indicate that the improvements in control do not compromise musical quality. This framework shows potential for future explorations in bar-level attribute control, with possible extensions to melody trends and other nuanced musical elements.

**Strengths:**

The technical approach, experimental design, and research problem are all well-motivated, with significant originality in the methodologies described in Sections 4.2 and 4.3. The paper addresses a meaningful challenge in music generation by achieving nuanced bar-level control without degrading the musical quality of outputs, providing value for both musicians and AI researchers in the field.

**Weaknesses:**

* The paper’s selection of baselines is somewhat limited. For example, models like ChatMusician also offer chord control, and waveform-based models such as Instruct-MusicGen could accomplish similar tasks. If the purpose is to enable the capability of following given chord/rhythm instruction, then comparing with more similar model are really useful.
* The evaluation covers only chord and rhythm intensity control without exploring melody control at the bar level. There are other elements which are essential to music that are worth observing.
* the assumption that music consists of chords and rhythm somehow limits the generalizability to broader music genres and styles. For example, bach's fuga are also usually used for symbolic music model development.
* The evaluation metrics are also not clearly justified; it remains unclear whether the task is multi-class or multi-label and why accuracy alone is used. Previous works in instruction-based music generation employ diverse metrics, such as Chroma cosine similarity for melody [1-2],  chord recognition accuracy [3-4], tempo bin accuracy [4], beat F1 score [2],  dynamics correlation [2], pitch-class histogram entropy [5], and similarity error [6], which could provide more comprehensive insights into the generated music quality.

[1] Jade Copet, Felix Kreuk, Itai Gat, Tal Remez, David Kant, Gabriel Synnaeve, Yossi Adi, and Alexandre Defossez. Sim- ´ ple and controllable music generation. Advances in Neural nformation Processing Systems, 36, 2024.
[2] Shih-Lun Wu, Chris Donahue, Shinji Watanabe, and Nicholas J Bryan. Music controlnet: Multiple time-varying controls for music generation. IEEE/ACM Transactions on Audio, Speech, and Language Processing, 2024.
[3] Bing Han, Junyu Dai, Xuchen Song, Weituo Hao, Xinyan He, Dong Guo, Jitong Chen, Yuxuan Wang, and Yanmin Qian. InstructME: An instruction guided music edit and remix framework with latent diffusion models. arXiv preprint arXiv:2308.14360, 2023.
[4] ] Jan Melechovsky, Zixun Guo, Deepanway Ghosal, Navonil Majumder, Dorien Herremans, and Soujanya Poria. MusTango: Toward controllable text-to-music generation. arXiv preprint arXiv:2311.08355, 2023.
[5] Shih-Lun Wu and Yi-Hsuan Yang. The jazz transformer on the front line: Exploring the shortcomings of ai-composed music through quantitative measures. In Proc. ISMIR, 2020.
[6] Botao Yu, Peiling Lu, Rui Wang, Wei Hu, Xu Tan, Wei Ye, Shikun Zhang, Tao Qin, and Tie-Yan Liu. Museformer: Transformer with fine-and coarse-grained attention for music 88 generation. Advances in Neural Information Processing Systems, 35:1376–1388, 2022

**Questions:**

* Section 4.1 notes that sometimes loss decreases, yet the model output does not follow the instructions. Why does the loss decline in these cases, and are there case studies to illustrate this (e.g., where four chord instructions yield only partial accuracy but musically incoherent outputs)?
* In Section 4.4, how is it determined during inference if a sample accurately reflects intended bar-level attributes? Are multiple language models used for inference evaluation?
* In Section 5.1.1, could the authors clarify the “REMI-like representation”?

**Details Of Ethics Concerns:**

The paper does not address potential copyright issues for the music datasets used in the experiments. Additionally, there is no mention of ethical review approval or participant demographics if musicians were involved in subjective evaluations.

---

### Official Review · Reviewer_puqK · 2024-11-03

**Soundness:** 1
**Presentation:** 2
**Contribution:** 2
**Rating:** 3
**Confidence:** 4

**Summary:**

This work proposes a training methodology aimed at improving the effectiveness of fine-grained bar-level control signals (i.e., additional tokens) in autoregressive symbolic music models. The authors identify issues with fine-tuning the MuseCoco model with added bar-level control signals on the POP909 dataset, and propose two methods to address these issues:

1. Adding an additional training objective in an effort to help MuseCoco to recognize the added control signals. The authors implement this by 'pre-adapting' the LoRA adapters to classify whether the bar-level control signals correspond to the musical content of the bar.

2. Adding an additional term to the loss function in an effort to promote the model to make different predictions when control signals are randomly replaced.

The authors evaluate their approach with objective metrics, such as bar-level accuracy of adherence to chords and rhythmic intensity, as well as with a subjective listening test with human participants. In both cases, the authors report that their approaches perform better than the baseline.

**Strengths:**

The main takeaway from this paper is that including the auxiliary training task described in (1) improves the model's performance in adhering to the bar-level control signals introduced at fine-tune time. The results of this experiment, shown in Table 3, demonstrate that adding pre-adaptation (1) improves accuracy, whereas adding the contrastive loss (2) has minimal impact.

The problem is well motivated in Sections 1 and 2, and the authors employ techniques that, to my knowledge, haven't been used in the context of controlling models for symbolic music generation.

**Weaknesses:**

There are several weaknesses in this paper that make me uncomfortable recommending it for acceptance.

- Novelty: The main successful approach described in the paper (1) has a long history of being applied in NLP as an auxiliary pre-training objective. Specifically, a variation of this approach was present in the training objective for BERT [1]. Although this contribution is novel in the context of symbolic music research, it limits the potential impact and novelty outside of this subdomain.

- Clarity: There are several sections of this paper that I found difficult to understand. I've highlighted some of the more significant ones in the questions section.

- Experimental Concerns: I have doubts about whether the subjective evaluation described in 5.1.5 is an accurate assessment of musicality, especially given the narrow margins and the similarity between the samples provided in the supplementary material. The subjective evaluation also doesn't appear to be relevant to the research hypotheses. In the objective evaluations described in 5.1.4, there is no explanation of how accuracy in chord adherence and rhythmic intensity are calculated. The comparison results with MuseCoco displayed in Table 1 appear to be quite marginal, which doesn't align with the claims made in L405-409. Using the sampling method described in 4.4 makes it difficult to analyze the objective evaluations, as this approach seems to force the model to adhere to the bar-level controls. Using K=1 (as is standard) would make for a more objective examination of the approach.

There are also some minor grammatical and referencing issues that could be improved. Some actionable items:

L40: I believe the OpenAI GPT-4 reference might be incorrect. Do you mean to reference the MuseNet [2] project by OpenAI?

L45: I believe references like this should be done with `/citet` as per the ICLR formatting guidelines.

L339: was -> were.

L344: invert -> inverse.

[1] Devlin, J., Chang, M.W., Lee, K. and Toutanova, K.N., 2018. BERT: Pre-training of Deep Bidirectional Transformers for Language Understanding. Proceedings of the 2018 Conference of the North American Chapter of the Association for Computational Linguistics: Human Language Technologies. Available at: https://arxiv.org/abs/1810.04805.

[2] Payne, Christine. "MuseNet." OpenAI, 25 Apr. 2019, openai.com/blog/musenet

**Questions:**

1) L191: What are the y_i?

2) L204-231: While I understand this takeaway, I wonder if the issue is simply insufficient fine-tuning data. Did you try fine-tuning on any other larger datasets? I have doubts that the reason for dataset selection given in lines 321-323 provides strong enough motivation. Testing the main hypothesis more rigorously could improve the strength of this work.

3) Section 4.4. I had difficulty parsing this subsection. What temperature is used for sampling? From my understanding, you are excluding tokens that don't adhere to the bar-level control signals. If this is the case, why bother with resampling at all? Instead, couldn't you simply impose logit-level controls?

4) Section 4.4. How do you measure whether a sample accurately reflects the bar-level attributes for rhythmic intensity and chord adherence? This is not clear to me, and chords have different functions in different musical contexts. This explanation is also missing from the experiments in Section 5.

5) L340: What is neta in this context?

---

### Official Review · Reviewer_jHTr · 2024-11-04

**Soundness:** 2
**Presentation:** 3
**Contribution:** 2
**Rating:** 3
**Confidence:** 4

**Summary:**

This paper addresses the challenge of achieving fine-grained, bar-level control in symbolic music generation, which is often limited in pretrained models. The authors propose a pre-training task for control initialization and a counterfactual loss function to address the challenge and yield a good improvement.

**Strengths:**

The topic of fine-grained, bar-level control in symbolic music is interesting and significant.

The presentation is clear.

The design of the loss function is insightful.

**Weaknesses:**

The contribution is limited. Based on the authors' description in the paper, this work is a bar-level controllable version of MuseCoco, adding bar-level chords, rhythm, and a new loss function to handle the newly added attributes. Additionally, the claim that "We conducted the first study in achieving fine-grained control of symbolic music generation based on the existing foundation model" is overstated, as many works achieve fine-grained control in symbolic music generation. For example, Theme Transformer generates music based on a given theme, a form of fine-grained control. WHOLE-SONG HIERARCHICAL is also a type of fine-grained control.

The experiment is incomplete. The authors emphasize that they propose solutions for effective updating of pretrained models with newly introduced control tokens in the title, abstract, and introduction. However, the experiment only shows refinement on one model, MuseCoco, which is insufficient to validate this claim. More experiments on additional pretrained models are necessary.

The related work on text-driven methods is not very relevant to this study, as this method does not involve any text modality.

**Questions:**

Since REMI is used, how do the authors measure global attributes such as time signature, instrument, and tempo?

Based on the distribution in Table 7, is it possible for an imbalance problem to arise? Can the model accurately recognise all chord intensities?

Could you further explain how K is used in inference? Does the model generate each bar at most K times to check if the bar meets the specified requirements?

Since chords generally follow music theory, why not use a rule-based method to enforce specific chords during inference, achieving 100% accuracy?

The paper emphasizes "effective updating of pretrained symbolic music models," which suggests a general method applicable to different models. Why do the authors focus only on MuseCoco? How might it be applied to other models, such as MusicTransformer, Compound Word Transformer, or ThemeTransformer? Additionally, I am interested in applications for models trained on large datasets, like Lakh.


In line 187, does |Xi| indicate the number of attributes or tokens? In your case, you include chords and rhythm, so this would be 2 if the bar has one chord and one rhythm intensity. Is token 24 just the value of xi,1? Should the sequence X be bolded to indicate it is a vector, not just a value?


What is X’, is it {X1, X2, ….Xb}, In line 253, are you modifying the bar-level attribute or the actual music sequence y? If either is modified, the notation in the equation should differ. In Equation 4, are you intending to indicate the probability of si=1 and sj=1. And is the function bar-level, calculated at each bar? Please clarify the formula and its explanation.

---

### Official Review · Reviewer_k9is · 2024-11-29

**Soundness:** 2
**Presentation:** 3
**Contribution:** 2
**Rating:** 3
**Confidence:** 5

**Summary:**

This paper suggests a methodology for bar-level controlled symbolic music generation. The methodology consists of a pre-training task imposing control prompts and a counterfactual loss to further guide the control adherence.

**Strengths:**

Presentation of the material is clear.

**Weaknesses:**

Contribution of the presented methodology is limited. Given its complexity, the approach is an overkill for the defined problem (bar-level controlled symbolic music generation, experimentally for chord control and rhythm intensity). There is not much emphasis on the music itself. Also, the definition of chord control is vague. In the evaluation, baselines are lacking as they do not include other fine-grained controlled symbolic music generation studies (see VAE-based literature as well).

**Questions:**

I strongly suggest that other fine-grained control literature should be considered for the baselines and the soundness of the presented approach should be justified, as apparently there is a significant gap between the complexity of the problem and the presented methodology.

---

### Note · Authors · 2025-01-09

I have read and agree with the venue's withdrawal policy on behalf of myself and my co-authors.